



# Ideas and perspectives: Emerging contours of a dynamic exogenous kerogen cycle

Thomas M. Blattmann[1]

[1]Biogeochemistry Program, Japan Agency for Marine-Earth Science and Technology (JAMSTEC), 2-15 Natsushima-cho, 237-0061 Yokosuka, Japan

*Correspondence to*: Thomas M. Blattmann (blattmannt@jamstec.go.jp)

**Abstract.** Growing evidence points to the dynamic role that kerogen is playing on the Earth's surface in controlling atmospheric chemistry over geologic time. Although quantitative constraints on weathering of kerogen remain loose, its changing weathering behavior modulated by the activity of glaciers, suggest that this largest pool of reduced carbon on Earth may have played a key part in atmospheric $CO_2$ variability across recent glacial-interglacial times and beyond.

## 1 Introduction

Over geologic timescales, atmospheric $CO_2$ is controlled by the combined effect of chemical weathering of silicates and carbonates and the organic carbon cycle (Berner, 1990; Blattmann et al., 2019a; Torres et al., 2014). Organic carbon in the form of kerogen comprises around 15 million PgC encompassing over 99.9% of reduced carbon present on Earth. Tectonic uplift and denudation subjects 150 PgC/kyr of kerogen to weathering on the Earth's surface thereby facilitating entry of this geologically ancient carbon into the atmosphere and other surficial carbon pools standing in close communication (Hedges and Oades, 1997). Upon oxidation of kerogen, $O_2$ is consumed and $CO_2$ is released to the atmosphere. In reverse, biospheric carbon burial in marine sediments removes carbon from the Earth's surface thereby drawing down atmospheric $CO_2$ and increasing $O_2$ over geological timescales (Galy et al., 2008). Therefore, kerogen weathering by uplift and sedimentary organic matter burial in the ocean play compensational roles in governing atmospheric chemistry (Fig. 1).

However, the decay of kerogen on the Earth's surface is incomplete, with physical erosion followed by riverine transport (Galy et al., 2015) and reburial in lacustrine and marine settings (Blattmann et al., 2018a; Blattmann et al., 2019b). The operation of this "simple cycle besides the more complicated common circulation of carbon", enunciated by Sauramo (1938), begs the questions (i) what is the reburial efficiency of kerogen, (ii) what are its controlling factors, and (iii) what are the implications of it changing for atmospheric chemistry over geologic timescales?

## 2 Kerogen and glaciers

Of the approximately 150 PgC/kyr of kerogen reaching the Earth's surface (Hedges and Oades, 1997), $43^{+61}_{-25}$ PgC/kyr is currently exported by rivers to oceans (Galy et al., 2015), indicating that the modern day reburial efficiency of this carbon lies



in the ballpark of 30% (10-70%). Paillard (2017) shows that changes in atmospheric $CO_2$ content and its stable carbon isotopic
composition over the course of recent glacial-interglacial cycles (Schmitt et al., 2012) is attributable to changes in the organic
carbon cycle. Here, shifting kerogen reburial efficiency presents itself as a candidate able to exert this $CO_2$ variability. Beyond
identifying a plausible source for increasing atmospheric $CO_2$ into interglacial times, evidence accumulated in over a century
of scientific studies supports that the reburial of kerogen has been more extensive during cold interludes in Earth history where
glacial erosion and ice rafting dominated (Blattmann et al., 2018b). This enhanced reburial efficiency strengthens the short-
circuiting of the exogenous kerogen cycle keeping this ancient reduced carbon locked away. However, under glaciofluvial
conditions, the entry of kerogen-bound carbon into surficial carbon pools is promoted (Horan, 2018), with, on the one hand,
glacial meltwater releasing dissolved organic matter from kerogen, which is readily consumed by microbes (Hood et al., 2009;
Schillawski and Petsch, 2008). Additionally, frost shattering together with the retreat of glaciers exposes finely ground, reactive
sediment, thereby accelerating oxidation and release of kerogen-bound carbon to the atmosphere (Horan et al., 2017). The
initially strong input of kerogen-derived $CO_2$ would die down as the availability of glacially ground, reactive kerogen declines
into the interglacial. Enhanced degradation of kerogen in the wake of glacial episodes is consistent with observations from
areas of ongoing deglaciation (Horan et al., 2017). Analogously, glaciers as agents for accelerating chemical weathering by
increased sediment yield and creation of high surface area reactive substrate have also been invoked for silicate and carbonate
minerals (Torres et al., 2017; Vance et al., 2009), with carbonate weathering constituting a source of $CO_2$ to the atmosphere
when sulfuric acid is involved (Torres et al., 2014). However, kerogen oxidation, owing to its faster weathering kinetics and
direct conversion to $CO_2$ (Horan et al., 2017), is a manifest process by which atmospheric $CO_2$ can be closely linked to glacial-
interglacial cyclicity (Fig. 2).

As kerogen is millions of years old, its oxidation releases radiocarbon-dead $CO_2$ to the atmosphere. Such a long-term decline
in the radiocarbon concentration of atmospheric $CO_2$ parallel to an overall increase in $CO_2$ amount did occur since the Last
Glacial Maximum (Reimer et al., 2013; Roth and Joos, 2013), consistent with such a dilutional process. In addition, the
response in atmospheric $CO_2$ faithfully echoing increasing global temperatures (Sigman and Boyle, 2000; Stips et al., 2016)
driven by orbital forcing (Hays et al., 1976) fits with enhanced oxidation of kerogen. Furthermore, across multiple glacial-
interglacial cycles, enhanced kerogen oxidation is also consistent with declining atmospheric $O_2$ over $10^6$ yr timescales (Stolper
et al., 2016).

## 3 Tackling geologic deep time

While modern day kerogen reburial efficiency is only loosely constrained, even less is known about how it varied back in
geologic time. Overarching controls on this process include the mode of erosion and transport ranging from glacial to
glaciofluvial to fluvial, remineralization intensity, controlled by continental margin type and geomorphology, and the intrinsic
reactivity of the kerogen present locally (Blair and Aller, 2012; Blattmann et al., 2018b). For the Paleocene-Eocene Thermal
Maximum, an extreme greenhouse episode, evidence increasingly points towards enhanced remineralization and leaching of





kerogen (Boucsein and Stein, 2009; Lyons et al., 2019), possibly enhanced by the activity of microbes (Hemingway et al., 2018; Petsch et al., 2001), thereby increasing the flux of carbon entering actively circulating pools on the Earth's surface. Across Earth's history over $10^9$ year timescales, kerogen reburial efficiency presumably varied as a function of atmospheric $O_2$ content, with lower $O_2$ contents tied to higher reburial efficiency (Daines et al., 2017). Kerogen is surmised to have acted

as a major source of carbon to the atmosphere and as an "antioxidant" during the early rise of atmospheric $O_2$ (Daines et al., 2017; Kump et al., 2011).

In order to understand atmospheric chemistry through geologic time, in addition to comprehensively budgeting the effect of mineral chemical weathering (Blattmann et al., 2019a; Hilton et al., 2014), changing reburial efficiency of sedimentary kerogen needs to be evaluated. Direct quantification of kerogen found reburied in sediments is often associated with considerable

uncertainty owing to uncertainties in organic matter source apportionment (Blattmann et al., 2019b) and geospatial variability (Blattmann et al., 2018a). While radiocarbon was paramount for establishing the importance of and quantifying kerogen reburial in the Recent (Blattmann et al., 2018b), the utility of radiocarbon quickly diminishes for strata preceding the Last Glacial Maximum owing to its radioactive decay. However, associated with kerogen are a promising suite of trace elements and their respective isotope signatures including, among others, rhenium (Hilton et al., 2014; Horan et al., 2017), osmium

(Georg et al., 2013; Ravizza and Esser, 1993), and iodine (Moran et al., 1998), that can be exploited to trace sedimentary kerogen and its degradation. In the case of osmium, seawater records reveal isotopic shifts at the beginning of interglacials attributable to the oxidation of kerogen (Georg et al., 2013), consistent with the presented hypothesis. More research constraining the exogenous kerogen cycle by quantification of reburied kerogen inputs (e.g. iodine isotopes, organic petrology) and kerogen oxidation recorded by chemical weathering proxies (e.g. osmium isotopes) is needed to put this hypothesis to the

test.

## 4 Outlook

As a cogwheel operating under manifold feedbacks in the greater Earth system (Sigman and Boyle, 2000), continuous glacial retreat and oxidation of finely ground kerogen provides a hypothesis consistent with a $CO_2$ increase in the wake of glacial episodes. In addition to studies of weathering in glacier forefields and source-to-sink tracing of sedimentary kerogen, several

lines of geochemical evidence including atmospheric carbon isotopic composition ([13]C and [14]C), which have thus far received contorted, partial explanations (Broecker and Clark, 2010; Schmitt et al., 2012), seawater osmium isotope changes, and long-term atmospheric $O_2$ content, conceptually go hand-in-hand with a simple opening and closing of the exogenous kerogen cycle modulated by glacial activity.

While basic controls on kerogen reburial efficiency have emerged, its quantitative impact on atmospheric chemistry through

geologic time remains conjectural. However, further study of pivotal episodes such as the Paleocene-Eocene Thermal Maximum under this lens may provide an outlook for geological processes relevant today. In the context of our warming



world, once critical thresholds are breached (Steffen et al., 2018), enhanced opening of the exogenous kerogen cycle carbon may put the Earth system on a trajectory that will possibly influence Earth's carbon cycle and climate for millennia to come.

**Acknowledgments**

This work was supported by funding from JAMSTEC. This work benefitted from discussions with Timothy Eglinton, Dominik Letsch, Maarten Lupker, Jesper Suhrhoff, Valier Galy, and Robert Hilton.

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

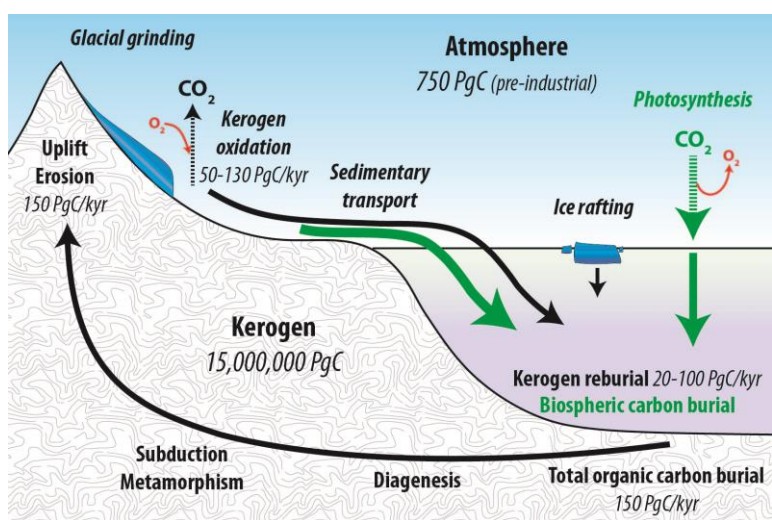


**Figure 1: Organic carbon cycle with the flow of kerogen (black solid lines) and the flow of biospheric carbon (green solid lines) showing both the fixation of atmospheric $CO_2$ by terrestrial and marine primary productivity. The combined flux of reworked kerogen and biospheric carbon into ocean sediments constitutes total organic carbon burial entering the endogenous kerogen pool (Galy et al., 2015; Hedges and Oades, 1997).**




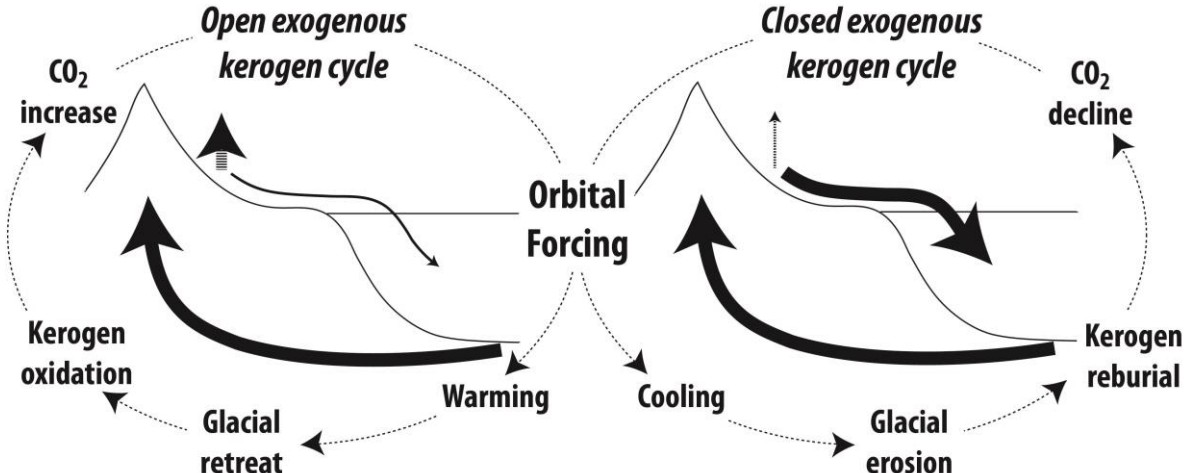

**Figure 2: Changes in kerogen reburial efficiency and its effect on the reentry of ancient carbon into surficial carbon pools as a function of weathering regime. In postglacial times, the oxidation of kerogen is more efficient leading to the exhalation of this carbon to the atmosphere. During glacial times, kerogen reburial is promoted by the activity of glaciers and ice sheets with relatively little**
**escape of this carbon during its transit across the Earth's surface.**