# Peer review of "Ideas and perspectives: Emerging contours of a dynamic exogenous kerogen cycle"

_Biogeosciences, 2019_

## Referee Comment (RC1) · Anonymous Referee #1 · 16 Sep 2019

Title of manuscript: Ideas and perspectives: Emerging contours of a dynamic exogenous kerogen cycle MS No.: bg-2019-273

MS Type: Ideas and perspectives

Author: Thomas Blattman

Recommendation: Reject

The manuscript entitled "Ideas and perspectives: Emerging contours of a dynamic exogenous kerogen cycle" by Blattman summarises the role that kerogen oxidation versus kerogen burial plays in balancing atmospheric $CO_2$ versus $O_2$. Some factors governing the transfer of organic carbon locked in kerogen from lithospheric storage to the atmosphere or offshore to be reburied are reported. Although the content is

interesting it only offers a brief review of the literature. As it stands, I consider the manuscript to be of insufficient detail for a review article. Furthermore, because it contains no new data or analysis it is unlikely to be of great value to the biogeosciences community. It does not really add much in terms of 'ideas and perspectives' to what is already known.

The abstract does not explain the purpose of this article or its novelty. It is also misleading, as the weathering of kerogen is not only modulated by the activity of glaciers (consider erosion, temperature, precipitation and so on, which have previously been discussed in the literature). It is also not clear to me why the author focuses on glacial-interglacial atmospheric $CO_2$ budgets when previous work (e.g. Petch, 2014; Bolton C cycle papers) outline the million year (rather than kyr) timeframe over which the kerogen cycle is relevant. Petch (2014) is an important review paper on organic carbon weathering that has not been cited

In the main text, the author postulates the role of kerogen reburial efficiency as a major contributing factor to atmospheric C budgets, referencing river basin data from Galy et al. (2015). In section 2, it would be worth pointing out that we do not have good constraint on how geochemical carbon fluxes change over time in large river basins globally. Furthermore, some consideration of weathering efficiency would be appropriate here. Although kerogen reburial efficiency may change over time, how too do weathering efficiency and weathering flux change. For example, high weathering intensity (low reburial of kerogen) may associated with low weathering fluxes, while lower weathering intensity (high kerogen reburial) is probably associated with high weathering fluxes. To what extent does weathering efficiency and weathering flux vary across the inorganic versus organic carbon cycles? Together, these ideas are relevant for achieving an integrated perspective of C cycling at Earth's surface through time. Although kerogen reburial efficiency may be an important driver of atmospheric $CO_2$ budgets, and may make sense in the context of data published in Reimer et al., 2013 and Roth and Joos, 2013, in glacial episodes there is also a large shift in the biospheric C distribution and

activity that would be critical to consider.

The idea of looking into kerogen reburial efficiency in the context of major climatic events (e.g. PETM) using trace elements is an interesting idea that could perhaps form the basis of a future research proposal. This article may therefore be more suitable as the basis of a proposal introduction. Alternatively, it could potentially be rewritten as a more in-depth and comprehensive review article.

I find figure 2 misleading with the terms 'open' and 'closed'. Presumably, the author is just trying to represent reduced kerogen reburial in periods of warming relative to cooler episodes?

Please also note the supplement to this comment:
https://www.biogeosciences-discuss.net/bg-2019-273/bg-2019-273-RC1-supplement.pdf

---

## Author Comment (AC1) · 18 Sep 2019

Dear Reviewer,

Thank you for your feedback and input. Ideas and perspectives articles in Biogeosciences "report new ideas and novel aspects of scientific investigations within the journal scope. Manuscripts of this type should be short (a few pages only)." This contribution is not written as nor intended as review. Other publications (e.g. Petsch, 2014) already serve this purpose. This work presents a simple and bold hypothesis of broad significance for atmospheric chemistry and the organic carbon cycle that is supported by the synthesized literature referenced within.

Yes, previous work (e.g. Petch, 2014; Bolton et al., 2006) discusses the million-year

timeframe over which the kerogen cycle is relevant. The refocusing on kyr-atmospheric $CO_2$ variability with emphasis on the Ice Ages and their possible connection with kerogen oxidation is a new idea which has thus far not been enunciated in the literature. This important aspect was not properly expressed in the manuscript and will be stated in the revised version for readers to understand the significance and novelty of the presented "Ideas and perspectives".

Previous work does consider erosion, temperature, precipitation etc. (e.g. Bolton et al. (2006)), and yes, will be brought up in the revised manuscript as well. However, recent insights highlighting the dynamic role of glacial activity on kerogen oxidation (e.g. Horan et al., 2017) and kerogen reburial (see references in Blattmann et al., 2018) make this an exciting possibility for explaining atmospheric $CO_2$ increase in the wake of waning glacial episodes affecting vast areas of high latitude area as outlined in this contribution. Figure 2 will be edited to make the hypothesis illustration clear in its meaning with respect to "open" and "closed" exogenous kerogen cycle.

Discussion of weathering efficiency and intensity are evidently key aspects in kerogen oxidation. While it is somewhat difficult to follow your (the reviewer's) exact line of thinking (i.e. distinguishing between physical and chemical weathering), chemical weathering efficiency (i.e. oxidation) of kerogen is greatly increased in glacial forefields with high surface area substrate left behind by glacial grinding and further fining driven by frost shattering (Horan et al., 2017). Alternatively, kerogen can also be incorporated in biomass by way of microbial activity consuming this fossil carbon (Petsch, 2014). The apportionment between fossil carbon entering the atmosphere directly or by first transiting the biosphere is subject of ongoing debate and complicates this discussion. In this context, kerogen reburial efficiency is a simple conceptual metric to view the kerogen cycle and its relevance for atmospheric chemistry, as kerogen that is reburied has no communication with (relatively) rapidly exchanging surficial carbon pools and therefore no effect on atmospheric $CO_2$. Ultimately, seawater osmium isotope signatures (and other trace elements) may provide a key constraint in quantifying kerogen

reburial on a global scale with the amount of exhumed carbon contained in kerogen expressed as the sum of carbon in reburied kerogen and degraded kerogen. If kerogen exhumation is approximated as constant, osmium isotopes (and other trace elements) can then be used to constrain global levels of kerogen decay on the Earth's surface through time, thereby giving us the key to kerogen reburial.

The question as to what extent weathering efficiency and weathering flux vary across inorganic and organic carbon cycles is a key question that needs to be addressed if we are to achieve a holistic understanding of the carbon cycle. To this end, Blattmann et al. (2019) and Horan et al. (2019) have presented regional studies from Taiwan and the Mackenzie River, respectively, that provide integrated carbon budgets based on chemical weathering of silicates, carbonates and quantitatively distinguish between sulfuric and carbonic acid weathering and combine these with estimates of kerogen oxidation. More such studies (to the knowledge of myself there are only two such regions in modern-day geologic space that are so rigorously characterized) are needed but is clearly beyond the scope of this "Ideas and perspectives" article (newly appeared Horan et al. 2019 will also be cited to the revised manuscript). However, yes, going back in time we need more constraints on these chemical weathering pathways and their fluxes. In this context, I argue that kerogen oxidation is a particularly dynamic component that appears promising to lay strong future research focus.

In this context, I would also like to add reference to Zeng (2003) who proposed the "Glacial Burial Hypothesis", which contains parallels to the hypothesis presented in this contribution. Zeng (2003) proposes the release of ice sheet-covered soil organic carbon in the wake of glacial episodes. One main problem that the idea faces is the bulldozing activity of glaciers leading to diminished presence of such soil organic carbon preserved under the ice (see Zeng (2007)). Overall, the modeling results of Zeng (2003, 2007), if soil organic carbon is reinterpreted as kerogen, lend further support to the plausibility to the idea of kerogen oxidation driving a substantial part of atmospheric $CO_2$ increase in the wake of glacial episodes. However, in the hypothesis

presented here, the bulldozing effect of ice and erosion of bedrock would produce a large supply of glacially ground rock, with kerogen contained within subject to oxidation once transported to the glacial forefront or more quickly exposed and "defrosted" during glacial retreat, thereby enhancing the overall effect. Another "alternative" land-based hypothesis is discussed in the volcanic degassing hypothesis (e.g. Sternai et al., 2016), which although similar in some respects (e.g. dilution of atmosphere with 14C-dead $CO_2$), is unable to account for the precise response of atmospheric $CO_2$ to orbital forcing (see Roth and Joos, 2012). Ocean-based hypotheses suffer from complicated, contorted explanations that cannot fully account for variations in carbon isotopic composition (13C and 14C) of atmospheric $CO_2$ over glacial-interglacial cycles (Broecker and Clark, 2010; Schmitt et al., 2012). Overall, the simple and, in my opinion, elegant hypothesis presented here appears well-positioned to explain a significant part of the enigmatic $CO_2$ variability across glacial-interglacial cycles.

This work enunciates the possibility of kerogen oxidation as a major driver in atmospheric $CO_2$ increase in the wake of glacial episodes. This hypothesis of kyr-timescale-relevance for this chemical weathering pathway is substantiated by several lines of independent evidence synthesized in this contribution including $CO_2$ carbon isotopic composition (13C and 14C), timing of $CO_2$ increase, seawater osmium record, kerogen oxidation studies, observations of kerogen reburial, and modeling results presented in other studies. Furthermore, bringing together the currently very small body of pioneering literature that has begun to sprout on this subject, a perspective is given on the relevance of kerogen oxidation for atmospheric $CO_2$ variability in the deep geologic past. One common denominator appears to emerge: the contours of a dynamic exogenous kerogen cycle!

I thank the reviewer for his/her time and effort and look forward to future discussions and strengthening the manuscript based on this input.

Sincerely,

Thomas Blattmann

18.09.2019 Yokosuka

References Blattmann, T. M., Letsch, D., and Eglinton, T. I., 2018, On the geological and scientific legacy of petrogenic organic carbon: American Journal of Science, v. 318, no. 8, p. 861-881. Blattmann, T. M., Wang, S. L., Lupker, M., Märki, L., Haghipour, N., Wacker, L., Chung, L. H., Bernasconi, S. M., Plötze, M., and Eglinton, T. I., 2019, Sulphuric acid-mediated weathering on Taiwan buffers geological atmospheric carbon sinks: Scientific Reports, v. 9, no. 1, p. 2945. Bolton, E. W., Berner, R. A., and Petsch, S. T., 2006, The weathering of sedimentary organic matter as a control on atmospheric O2: II. Theoretical modeling: American Journal of Science, v. 306, no. 8, p. 575-615. Broecker, W., and Clark, E., 2010, Search for a glacial-age 14C-depleted ocean reservoir: Geophysical Research Letters, v. 37, no. 13. Horan, K., Hilton, R. G., Selby, D., Ottley, C. J., Gröcke, D. R., Hicks, M., and Burton, K. W., 2017, Mountain glaciation drives rapid oxidation of rock-bound organic carbon: Science Advances, v. 3, no. 10. Horan, K., Hilton, R. G., Dellinger, M., Tipper, E., Galy, V., Calmels, D., Selby, D., Gaillardet, J., Ottley, C. J., Parsons, D. R., and Burton, K. W., 2019, Carbon dioxide emissions by rock organic carbon oxidation and the net geochemical carbon budget of the Mackenzie River Basin: American Journal of Science, v. 319, no. 6, p. 473-499. Petsch, S. T., 2014, Weathering of organic carbon, in Holland, H. D., and Turekian, K. K., eds., Treatise on Geochemistry: Oxford, Elsevier, p. 217-238. Roth, R., and Joos, F., 2012, Model limits on the role of volcanic carbon emissions in regulating glacial–interglacial CO2 variations: Earth and Planetary Science Letters, v. 329-330, p. 141-149. Schmitt, J., Schneider, R., Elsig, J., Leuenberger, D., Lourantou, A., Chappellaz, J., Köhler, P., Joos, F., Stocker, T. F., Leuenberger, M., and Fischer, H., 2012, Carbon isotope constraints on the deglacial CO2 rise from ice cores: Science, v. 336, no. 6082, p. 711-714. Sternai, P., Caricchi, L., Castelltort, S., and Champagnac, J.-D., 2016, Deglaciation and glacial erosion: A joint control on magma productivity by continental unloading: Geophysical Research Letters, v. 43, no. 4, p. 1632-1641.

Zeng, N., 2003, Glacial-interglacial atmospheric CO2 change — The glacial burial hypothesis: Advances in Atmospheric Sciences, v. 20, no. 5, p. 677-693. Zeng, N., 2007, Quasi-100 ky glacial-interglacial cycles triggered by subglacial burial carbon release: Climate of the Past, v. 3, no. 1, p. 135-153.

Please also note the supplement to this comment:
https://www.biogeosciences-discuss.net/bg-2019-273/bg-2019-273-AC1-supplement.pdf

---

## Referee Comment (RC2) · Anonymous Referee #2 · 19 Sep 2019

This short paper presents an interesting discussion on the role of the kerogen cycle onto the overall carbon cycle and the climate system. I basically agree with the author that it is an important piece of the puzzle. Such a short paper might therefore be a valuable contribution to the EGU journal Biogeosciences and I would encourage publication. Still, I mostly disagree with the author on the question of glacial-interglacial cycles. Perturbations in the kerogen fluxes are probably far too small to generate a 100 ppm atmospheric CO2 change over a few thousands of years. This can be explained through simple back-of-the-envelope calculations, something which is lacking in the current manuscript. I would therefore recommend some major changes in this direction.

Back-of-the-envelope calculations:

According to the author (eg. line 27), the kerogen flux is approximately 0.15 PgC/yr, with a reburial efficiency of the order of 30% (between 10% and 70%). The net flux might therefore be about 0.1 PgC/yr. The key question is to decide how large this flux might change through time, in particular across a glacial-interglacial transition. If major erosional changes are expected at high latitudes, this is certainly less true at lower latitudes, where much of the current weathering is taking place today. It seems therefore quite unlikely that glacial-interglacial changes would affect kerogen fluxes by more than 100%. Such an overestimated perturbation, that would persist during the whole duration of the last deglaciation, about 10 thousand years, would therefore change the Earth carbon budget by 1000 PgC. This would have two consequences: - First, it would raise the overall carbon content at the Earth surface (actually, mainly the ocean, with is today around 38000 PgC) by about 2,5%. The atmospheric part scales with the square of this overall content, and could therefore rise by about 5% or equivalently by about 10-15 ppm. Such an extreme hypothesis leads therefore to a change one order of magnitude smaller than the observed glacial-interglacial one, with a rise from 180 ppm at the last glacial maximum to about 280 ppm at pre-industrial times. - Second, and more importantly, it would lower significantly the Earth carbon $\delta13C$ signature. Assuming a kerogen $\delta13C$ of -25‰ this would translate into a change of about -0.6‰ of the ocean DIC $\delta13C$. But the observations from marine cores are well-known to show just the opposite signal, with a very significant increase of about +0.3‰ across all deglaciations, generally attributed to the re-growth of forest (Shackleton et al., 1977; Shackleton et al., 1983). In other words, we know that glacial-interglacial carbon transitions are not linked to a release of isotopically light carbon, but on the contrary are most probably associated with an increase of the organic carbon reservoir on Earth. Overall, these facts show that the "kerogen hypothesis" cannot explain the bulk of glacial-interglacial carbon cycle changes. In contrast there are many indications, both from paleoclimatology and from modeling experiments, that glacial-interglacial CO2 changes are tightly connected to Southern ocean processes. This does not mean that kerogens must be entirely overlooked, but they are obviously much more likely to be important on longer

time scales. In particular, when citing Paillard, 2017 (lines 29-31), the manuscript is somewhat misleading. In this paper, it is suggested that organic carbon does contribute significantly to the 400-kyr oscillations found in the carbon isotopic records, not specifically during the Quaternary, but also during many other time periods. The Quaternary context is used to assess consistency with available $CO_2$ and 13C data over the last few millions of years, when assuming that kerogens are involved in carbon and climate variability on time scales of a few hundreds of thousands of years (as detailed above, there is no such consistency for glacial-interglacial transitions). Furthermore it might be interesting to better understand how glacial erosion during the Quaternary affects kerogen fluxes when compared to the Pliocene or earlier periods. Such a million-year time scale is likely a much better framework for discussing the role of kerogen on $CO_2$ and climate.

References:

Paillard, D. (2017). The Plio-Pleistocene climatic evolution as a consequence of orbital forcing on the carbon cycle. Climate of the Past, 13, 1259–1267.

Shackleton, N. J., Andersen, N. R., Malahoff, A., & Gibbs, R. J. (1977). The fate of fossil fuel CO2 in the oceans. In The fate of fossil fuel CO2 in the oceans (Vol. 6, pp. 401–427).

Shackleton, N. J., Hall, M., Line, J., & Shuxi, C. (1983). Carbon isotope data in core V19-30 confirm reduced carbon dioxide concentration in the ice age atmosphere. Nature, 306, 319–322.

---

## Author Comment (AC2) · 1 Oct 2019

Dear Reviewer,

Thank you for your feedback and input. Overall, the boundary conditions imposed by proxy information collected over decades of work basically amount to one undeniable fact: we are unable to fully reconcile and align glacial-interglacial CO2 variability with changes in the biosphere and gas exchange with the ocean. We therefore need a new hypothesis.

On the most basic level, the constraints imposed by radiocarbon clearly indicate we are missing one fundamental parameter: a source of carbon depleted or devoid of radiocarbon (Broecker and Clark 2010; Zhao et al., 2018). When asking ourselves what

this source may be, the transition during times of highest CO2 increase between glacial and interglacial clearly reveals a negative stable carbon isotopic shift of atmospheric CO2 (Smith et al., 1999; Schmitt et al., 2012), which is a strong indicator of respired organic carbon acting as source to the atmosphere (Bauska et al., 2016).

With this, exogenous kerogen becomes an obvious candidate as the missing player in the global carbon cycle. It fulfills the basic requirements: it is abundant, it is isotopically light, and it is radiocarbon-dead. For an additional host of reasons, it appears plausible, as elaborated in the manuscript, with kerogen reburial efficiency higher during glacial episodes in Earth's history as a straightforward supporting observation (see references in Blattmann et al., 2018).

As pointed out by yourself (the reviewer), viewing the glacial and interglacial episodes as individual time intervals, one may draw the conclusion that there is biospheric uptake of isotopically light carbon on land (Shackleton 1977; Shackleton et al., 1983), which I agree, is most likely true. However, this view regards the two climate states as steady state conditions and does not consider transitional dynamics. Also, stable carbon isotopic changes are trickier to interpret as carbon isotopes fractionate as they rotate between different carbon pools changing in size; in contrast, radiocarbon isotope composition of atmospheric CO2, which is fractionation corrected, is more straightforward in its interpretation. Ultimately, biospheric activity and atmosphere-ocean exchange cannot account for the radiocarbon budget and its evolution from the Last Glacial Maximum to the present. However, yes, the kerogen cycle is one "cogwheel operating under manifold feedbacks in the greater Earth system" (original manuscript) and clearly this point needs to be expanded on to present a mediated view.

How important is this cogwheel? And you (the reviewer) rightfully points out that a back-of-the-envelope calculation is lacking. The "Glacial Burial Hypothesis" (Zeng, 2003), which contains parallels to the hypothesis presented here, takes care of this in a detailed manner. Zeng (2003) proposes the oxidation of ice sheet-covered soil organic carbon during glacial retreat. Overall, the modeling results of Zeng (2003, 2007), if

soil organic carbon is reinterpreted as kerogen, lend further support to the idea that kerogen oxidation drives a substantial part of atmospheric $CO_2$ increase in the wake of glacial episodes. In quantitative terms, Zeng (2003) attributes the oxidation of 547 PgC soil organic carbon to a 60 PgC increase in atmospheric $CO_2$ (increase of 30 ppm). I view these estimates as conservative, because unlike Ning Zeng's assumption of soil organic carbon oxidation, the supply of kerogen shed and subject to temporary storage on land over millennia of glaciation is much vaster! Overall, the dynamism of the exogenous kerogen cycle over the Ice Ages may seems further strengthened by increased bedrock exhumation (Herman et al., 2013; Herman et al., 2015), thereby leading to greater detrital fluxes of kerogen reburied into ocean sediments and also enhanced supply of ground kerogen exposed to the elements in the wake of glacial episodes.

I thank the reviewer for his/her time and effort and look forward to future discussions and strengthening the manuscript based on this input and adding the references cited herein.

Sincerely,

Thomas Blattmann

01.10.2019 Yokosuka

References

Bauska, T. K., Baggenstos, D., Brook, E. J., Mix, A. C., Marcott, S. A., Petrenko, V. V., Schaefer, H., Severinghaus, J. P., and Lee, J. E., 2016, Carbon isotopes characterize rapid changes in atmospheric carbon dioxide during the last deglaciation: Proceedings of the National Academy of Sciences, v. 113, no. 13, p. 3465-3470. Blattmann, T. M., Letsch, D., and Eglinton, T. I., 2018, On the geological and scientific legacy of petrogenic organic carbon: American Journal of Science, v. 318, no. 8, p. 861-881. Broecker, W., and Clark, E., 2010, Search for a glacial-age 14C-depleted ocean

reservoir: Geophysical Research Letters, v. 37, no. 13. Herman, F., Seward, D., Valla, P. G., Carter, A., Kohn, B., Willett, S. D., and Ehlers, T. A., 2013, Worldwide acceleration of mountain erosion under a cooling climate: Nature, v. 504, p. 423. Herman, F., Beyssac, O., Brughelli, M., Lane, S. N., Leprince, S., Adatte, T., Lin, J. Y. Y., Avouac, J.-P., and Cox, S. C., 2015, Erosion by an Alpine glacier: Science, v. 350, no. 6257, p. 193. Schmitt, J., Schneider, R., Elsig, J., Leuenberger, D., Lourantou, A., Chappellaz, J., Köhler, P., Joos, F., Stocker, T. F., Leuenberger, M., and Fischer, H., 2012, Carbon isotope constraints on the deglacial CO2 rise from ice cores: Science, v. 336, no. 6082, p. 711-714. Shackleton, N. J., 1977, Carbon-13 in Uvigerina: Tropical rainforest history and the equatorial Pacific carbonate dissolution cycles, in Andersen, N. R., and Malahoff, A., eds., The Fate of Fossil Fuel CO2 in the Oceans, Plenum Press, p. 401-427. Shackleton, N. J., Hall, M. A., Line, J., and Shuxi, C., 1983, Carbon isotope data in core V19-30 confirm reduced carbon dioxide concentration in the ice age atmosphere: Nature, v. 306, no. 5941, p. 319-322. Smith, H. J., Fischer, H., Wahlen, M., Mastroianni, D., and Deck, B., 1999, Dual modes of the carbon cycle since the Last Glacial Maximum: Nature, v. 400, no. 6741, p. 248-250. Zhao, N., Marchal, O., Keigwin, L., Amrhein, D., and Gebbie, G., 2018, A synthesis of deglacial deep-sea radiocarbon records and their (in)consistency with modern ocean ventilation: Paleoceanography and Paleoclimatology, v. 33, no. 2, p. 128-151. Zeng, N., 2003, Glacial-interglacial atmospheric CO2 change — The glacial burial hypothesis: Advances in Atmospheric Sciences, v. 20, no. 5, p. 677-693. Zeng, N., 2007, Quasi-100 ky glacial-interglacial cycles triggered by subglacial burial carbon release: Climate of the Past, v. 3, no. 1, p. 135-153.

Please also note the supplement to this comment:
https://www.biogeosciences-discuss.net/bg-2019-273/bg-2019-273-AC2-supplement.pdf